# Analysis of Trapping Effect on Large-Signal Characteristics of GaN HEMTs Using X-Parameters and UV Illumination

**DOI:** 10.3390/mi14051011

**Published:** 2023-05-08

**Authors:** Kun-Ming Chen, Chuang-Ju Lin, Chia-Wei Chuang, Hsuan-Cheng Pai, Edward-Yi Chang, Guo-Wei Huang

**Affiliations:** 1Taiwan Semiconductor Research Institute, Hsinchu 300091, Taiwan; yakoai@narlabs.org.tw (C.-J.L.); cwchuang@narlabs.org.tw (C.-W.C.); 2International College of Semiconductor Technology, National Yang Ming Chiao Tung University, Hsinchu 300093, Taiwan; edc@nycu.edu.tw; 3Institute of Electronics, National Yang Ming Chiao Tung University, Hsinchu 300093, Taiwan

**Keywords:** GaN, HEMT, large-signal characterization, trapping effect, UV illumination, X-parameter

## Abstract

GaN high-electron-mobility transistors (HEMTs) have attracted widespread attention for high-power microwave applications, owing to their superior properties. However, the charge trapping effect has limitations to its performance. To study the trapping effect on the device large-signal behavior, AlGaN/GaN HEMTs and metal-insulator-semiconductor HEMTs (MIS-HEMTs) were characterized through X-parameter measurements under ultraviolet (UV) illumination. For HEMTs without passivation, the magnitude of the large-signal output wave ($X_{21}^{\mathrm{FB}}$) and small-signal forward gain ($X_{2111}^{S}$) at fundamental frequency increased, whereas the large-signal second harmonic output wave ($X_{22}^{\mathrm{FB}}$) decreased when the device was exposed to UV light, resulting from the photoconductive effect and suppression of buffer-related trapping. For MIS-HEMTs with SiN passivation, much higher $X_{21}^{\mathrm{FB}}$ and $X_{2111}^{S}$ have been obtained compared with HEMTs. It suggests that better RF power performance can be achieved by removing the surface state. Moreover, the X-parameters of the MIS-HEMT are less dependent on UV light, since the light-induced performance enhancement is offset by excess traps in the SiN layer excited by UV light. The radio frequency (RF) power parameters and signal waveforms were further obtained based on the X-parameter model. The variation of RF current gain and distortion with light was consistent with the measurement results of X-parameters. Therefore, the trap number in the AlGaN surface, GaN buffer, and SiN layer must be minimized for a good large-signal performance of AlGaN/GaN transistors.

## 1. Introduction

GaN-based HEMTs have been widely investigated for next-generation wireless communication and power electronics applications [1,2,3,4]. Owing to the high breakdown voltage and high electron velocity, they exhibit excellent RF power performance at high frequencies. GaN HEMTs fabricated on sapphire substrate were demonstrated to be operated at 94 GHz with an output power density of 5.8 W/mm and a power-added efficiency of 38.5% [3]. Moreover, for the purposes of keeping costs low and integrating with Si-based devices, GaN-on-Si technology has also been developed [5,6]. A high maximum oscillation frequency (f_max_) of 270 GHz has been achieved for GaN-on-Si HEMT, suggesting its great application potential for millimeter-wave power circuits. However, the reliability issues related to gate leakage and current collapse degrades the output power and efficiency of GaN-based devices [7,8]. The current collapse phenomena are associated with the presence of charge trapping states at the surface, or in the GaN buffer [9,10]. One of the popular methods for reducing the gate leakage current is introducing a SiN gate dielectric layer under the metal gate. The SiN layer in this MIS-HEMT can also suppress the surface trap states [11]. Although the MIS-HEMT shows lower gate leakage and less surface charge trapping effect, it still suffers from buffer-induced current collapse. Furthermore, the existence of traps in SiN gate dielectric will result in threshold voltage instability [12].

In order to optimize the device RF power performance and build a large-signal model that takes the trapping effect into account, the charge trapping behavior of GaN-based HEMTs has been investigated through S-parameter and load-pull measurements [8,13,14]. Various large-signal models including the trapping-related components were proposed to simulate the device nonlinear characteristics. The common modeling methods are based on equivalent circuits with current and charge sources represented by closed-form equations [15,16,17]. The extraction of model parameters often relies on pulsed current–voltage (I–V) and multi-bias S-parameter measurements, and the model development is time consuming. For fast and efficient model extraction, measurement-based models utilizing a lookup table [18,19] or neural network techniques [20,21] have been developed to simulate the large-signal characteristics of GaN devices. Other measurement-based models using X-parameters were also presented in [22,23] for GaN device modeling, but they did not describe the charge trapping effect. X-parameters are based on the polyharmonic distortion (PHD) behavioral model and can be regarded as an extension of the well-known S-parameters under large-signal conditions. They capture the device behavior at the fundamental and harmonic frequencies in a single measurement for a given bias point. Therefore, the nonlinear microwave 2-port networks can be described by X-parameters. Recently, X-parameters have been utilized in device nonlinear characterization, large-signal modeling, and circuit design [22,23,24,25].

In this study, we investigate the influence of charge trapping on the large-signal characteristics of AlGaN/GaN HEMTs and MIS-HEMTs by analyzing X-parameters. In order to change the charge status of traps, a 365 nm UV LED source was used to illuminate the wafer during X-parameter measurements. Therefore, we can use the UV illumination method to examine the influence of different trap types on the device characteristics. The main contents are organized as follows. Section 2 describes the device structures used in this study and outlines the concept of X-parameters. In Section 3, the X-parameter measurement results of AlGaN/GaN HEMTs and MIS-HEMTs under UV illumination are demonstrated. The RF signal waveforms simulated based on the X-parameter model are also presented. Section 4 provides the conclusion.

## 2. Materials and Methods

The test devices used in this study were depletion-mode unpassivated Schottky-gate AlGaN/GaN HEMTs and passivated AlGaN/GaN MIS-HEMTs. The schematic cross sections of device structures are shown in Figure 1. The epitaxial layer was grown by metal-organic chemical vapor deposition (MOCVD) on 6-inch silicon substrate. It consisted of an unintentionally doped 3 μm thick GaN buffer and a 25 nm thick Al_0.22_Ga_0.78_N barrier layer. For MIS-HEMTs, an additional 25 nm thick SiN layer was deposited on top of the AlGaN barrier by plasma-enhanced chemical vapor deposition (PECVD) at 300 °C to passivate the surface trap states and serve as a gate dielectric. After mesa isolation, a Ti/Al/Ni/Au stack was formed by electron-beam evaporation and annealed at 800 °C for 60 s as source/drain ohmic contact. Finally, the Ni/Au gate electrode was evaporated and patterned by a metal lift-off process. The fabricated devices consisted of two gate fingers with a gate width of 2 × 25 μm and a gate length of 3 μm. The gate–source spacing (L_GS_) and gate–drain spacing (L_GD_) were 2.5 μm and 10 μm, respectively.

X-parameter measurements were performed using the Keysight nonlinear vector network analyzer (NVNA). The system is based on a Keysight N5242B PNA-X dual-source network analyzer with a phase reference comb generator for phase calibration and a NVNA application software for instrument control for highly automated X-parameter characterization and extraction. The expression for X-parameters used in this study is given by [26]
(1)$$b_{\mathrm{pm}}=X_{\mathrm{pm}}^{\mathrm{FB}}\left(\left|A_{11}\right|\right)P^{m}+\sum_{q,n}X_{\mathrm{pmqn}}^{S}\left(\left|A_{11}\right|\right)P^{m-n}a_{\mathrm{qn}}+\sum_{q,n}X_{\mathrm{pmqn}}^{T}\left(\left|A_{11}\right|\right)P^{m+n}a_{\mathrm{qn}}^{*}$$
where b_pm_ is the scattered wave, a_qn_ is the small-signal incident wave, q and p are the port numbers, n and m are the harmonic orders, A_11_ is the large-signal incident wave at port 1 with fundamental frequency, and P is the unit length phasor with the same phase as A_11_. The $X_{\mathrm{pm}}^{\mathrm{FB}}$, $X_{\mathrm{pmqn}}^{S}$, and $X_{\mathrm{pmqn}}^{T}$ parameters are characterized as a function of A_11_ and used to generate a lookup table model to provide the X-parameter representation of the device. $X_{\mathrm{pm}}^{\mathrm{FB}}$ represents the response of the device to a large input signal, while $X_{\mathrm{pmqn}}^{S}$ and $X_{\mathrm{pmqn}}^{T}$ depict the linear mapping of small-signal incident waves into scattered waves. The characterization was carried out by applying to each port and each harmonic the reference resistance of 50 Ω.

## 3. Results and Discussion

### 3.1. Measured X-Parameters of AlGaN/GaN HEMTs

Figure 2a displays the magnitudes of $X_{21}^{\mathrm{FB}}$, $X_{22}^{\mathrm{FB}}$, and $X_{23}^{\mathrm{FB}}$ versus input power for an AlGaN/GaN HEMT biased at gate voltage V_GS_ = −1.5 V and drain voltage V_DS_ = 20 V. This bias condition corresponds to the bias point for peak current-gain cutoff frequency (f_T_). X-parameters were measured at an excitation frequency of 1 GHz and the third-order harmonics were selected. As shown in this figure, the illuminated device exhibited higher $X_{21}^{\mathrm{FB}}$, lower $X_{22}^{\mathrm{FB}}$, and similar $X_{23}^{\mathrm{FB}}$ compared with the dark one. Since $X_{21}^{\mathrm{FB}}$, $X_{22}^{\mathrm{FB}}$, and $X_{23}^{\mathrm{FB}}$ are related to the fundamental, second harmonic and third harmonic output powers, respectively, these results indicate that the output power and linearity of HEMT have been improved under UV illumination. Because the photon energy of 365 nm UV light corresponds to the bandgap of GaN, the photons are only absorbed in the GaN buffer layer, generating electron–hole pairs in this region. The photo-generated electrons flow into the two-dimensional electron gas (2-DEG) channel and increase the drain current I_D_ (i.e., photoconductive effect), while the generated holes accumulate in the GaN buffer [27]. The photo-generated holes neutralize the trapped electrons in the buffer layer. The suppression of buffer-related charge trapping under UV illumination has been reported based on the pulse measurement method [28]. In previous literature [29], it was mentioned that the trapped electrons may also release from the surface state after gaining the photon energy, but this phenomenon was not noticeable in our device [30]. The photoconductive effect increases the 2-DEG concentration, which enhances the transconductance (g_m_) and reduces the source/drain access resistances. When electrons are captured by the traps near the gate edge in the GaN buffer, the 2-DEG concentration in the gate–drain access region is depleted, leading to a high drain resistance. The suppression of buffer-related charge trapping by UV light will therefore lower the drain resistance [30]. As a result, the illuminated device exhibits higher $X_{21}^{\mathrm{FB}}$ than the dark one. 

To explain the variations of $X_{22}^{\mathrm{FB}}$ and $X_{23}^{\mathrm{FB}}$ with UV light, we calculated the transconductance (g_m_), first g_m_ derivative (g_m_′) and second g_m_ derivative (g_m_″) from the measured I_D_-V_GS_ curves. The results are shown in Figure 2b. At V_GS_ = −1.5 V, the changes of g_m_, g_m_′, and g_m_″ with light are consistent with those of $X_{21}^{\mathrm{FB}}$, $X_{22}^{\mathrm{FB}}$, and $X_{23}^{\mathrm{FB}}$. Therefore, the linearity of these devices is highly affected by transconductance and its derivatives. For a low-distortion operation, the magnitude differences between fundamental and harmonic output powers should be as large as possible. For HEMTs under UV illumination, these differences are larger than the ones in the dark, indicating that the linearity has been improved by the photoconductivity effect and suppression of buffer-related trapping. 

Figure 3 shows the magnitudes of $X_{1111}^{S}$, $X_{2111}^{S}$, $X_{2121}^{S}$, and $X_{2121}^{T}$ of an AlGaN/GaN HEMT. We found that $X_{2111}^{S}$ decreases rapidly with increasing input power due to the trapping effect. This observation of gain compression cannot be carried out using conventional S-parameter measurements, since they only reveal the small-signal gain S_21_. When the device is exposed to UV light, the magnitude of $X_{2111}^{S}$ increases, indicating that the AlGaN/GaN HEMT will have a higher power gain under UV illumination. For the design of input and output matching networks of a power amplifier, $X_{1111}^{S}$, $X_{2121}^{S}$, and $X_{2121}^{T}$ of the power transistor are important parameters [31]. However, as depicted in Figure 3, the magnitudes of $X_{2121}^{T}$ in dark and light conditions are much smaller than $X_{2121}^{S}$, indicating that the output nonlinear behavior is insignificant. Therefore, in our case, the circuit design can ignore the response of the device to the conjugate input signal, and we only consider the values of $X_{1111}^{S}$ and $X_{2121}^{S}$ to optimize the source and load impedances, respectively. From Figure 3, we also observed the change in $X_{1111}^{S}$ and $X_{2121}^{S}$ under UV light, which means the input and output reflection coefficients at fundamental frequency have changed, owing to the reduced resistance of the channel, source, and drain regions [26]. From S-parameter measurements, we can obtain the channel, source, and drain resistances, which are 3533, 19, and 186 Ω for the HEMT in the dark, and change to 2681, 17, and 134 Ω under UV illumination, respectively. The reduction in channel resistance will reduce $X_{2121}^{S}$ and increase $X_{1111}^{S}$. The reduction in source resistance will further increase $X_{1111}^{S}$.

### 3.2. Measured X-Parameters of AlGaN/GaN MIS-HEMTs

The surface trap density in HEMTs can be reduced by depositing a SiN passivation layer upon the AlGaN barrier. This SiN insulating layer also acts as a gate dielectric to suppress gate leakage. The measured main X-parameters of the passivated AlGaN/GaN MIS-HEMT in the dark and under UV illumination are depicted in Figure 4. The test device is biased at the condition that the peak f_T_ is obtained. Compared with HEMT, MIS-HEMT shows higher $X_{21}^{\mathrm{FB}}$, and lower $X_{22}^{\mathrm{FB}}$ and $X_{23}^{\mathrm{FB}}$. It demonstrates that the device output power and linearity can be apparently improved by mitigating the surface charge trapping. However, the photo-induced $X_{21}^{\mathrm{FB}}$ enhancement in MIS-HEMT was not observed clearly when it was exposed to UV light. This is because excess charge trapping centers in the amorphous SiN layer are created by photons [32], which degrades g_m_ and thus offsets the benefit of the photoconductive effect. Even so, we still observed a decrease in $X_{23}^{\mathrm{FB}}$ under illumination due to the decrease in drain resistance caused by the suppression of buffer-related trapping.

As with the X^FB^ parameters, the X^S^ and X^T^ coefficients of the AlGaN/GaN MIS-HEMT also show less dependence on UV light, as illustrated in Figure 4b. It should be noted that the trend of $X_{1111}^{S}$ variation with UV light is opposite to that in HEMT. This result might be due to increased gate leakage caused by photo-induced traps in the SiN gate dielectric. The gate leakage can be denoted by a resistance in parallel with the gate-to-source capacitance. Higher leakage means lower resistance, resulting in a lower input reflection coefficient. From Figure 4b, we found many improvements in the properties of MIS-HEMTs compared to HEMTs due to the reduced surface charge trapping. Firstly, MIS-HEMT exhibits higher $X_{2111}^{S}$ and lower $X_{2121}^{T}$ coefficients than HEMT under both dark and light conditions, indicating it has higher power gain and better output linearity. Secondly, the $X_{2111}^{S}$ gain compression at high input power is not significant in MIS-HEMT. Finally, $X_{1111}^{S}$ and $X_{2121}^{S}$ do not vary with the power level, suggesting matching networks are easier to design when the transistor is used in power amplifier circuits.

### 3.3. Large-Signal Modeling Results

The large-signal characteristics of AlGaN/GaN HEMTs and MIS-HEMTs were simulated based on the X-parameter model using the ADS circuit simulation tool. The circuit configuration for large-signal simulation is shown in Figure 5a. The dc biases, input power (P_in_), and frequency are set to match the X-parameter measurement conditions. The source and load impedances (Z_S_ and Z_L_) are 50 Ω. Typical simulated power characteristics are shown in Figure 5b. The output powers measured by a load-pull system with 50 Ω load impedance are also displayed in this figure to verify the accuracy of the X-parameter model. It can be seen that the simulated curves agree well with the measured data; therefore, the X-parameter model can describe the device large-signal performance.

Figure 6 shows the gate and drain current waveforms in the time domain obtained by X-parameter model simulation. For HEMT devices, we observed significant current distortion at an input power above −10 dBm in the dark and it becomes more serious when devices operate at higher power. Moreover, the average drain current decreases with increasing RF power level, demonstrating that the distortion is caused by the trapping effect [15]. The drop in average current is a significant consequence of the asymmetric nature of the trap capture and emission processes. At high frequencies, when the signal period is lower than the charge emission time, the trap status cannot respond quickly enough to the change of the applied signal, leading to the compression of maximum current. For MIS-HEMTs, no significant current distortion was observed and the average current was barely affected by the power level, resulting from the mitigation of surface-related charge trapping. 

When the devices are exposed to UV light, several phenomena are observed in the current waveforms, owing to the increase in photocurrent and suppression of buffer-related trapping. (1) The average drain current in HEMT and MIS-HEMT increases. The light-enhanced drain current is consistent with the dc measurement result [30]. (2) The peak-to-peak amplitude of drain current, which is associated with $X_{21}^{\mathrm{FB}}$, for the HEMT under UV illumination is higher than that in the dark environment. (3) The current distortion in HEMT is improved under illumination. However, the improvement is minor due to the serious surface-related trapping still existent in the illuminated HEMT.

For MIS-HEMTs, the UV light will induce excess trap centers in the amorphous SiN layer, which offsets the performance improvement resulting from the photoconductive effect and the suppression of buffer charge trapping. Therefore, the peak-to-peak amplitude of drain current is not influenced by light. Furthermore, the photo-induced traps in SiN slightly increase the gate current of MIS-HEMTs.

## 4. Conclusions

In this work, we have investigated the impact of trapping effect on the large-signal characteristics of AlGaN/GaN HEMTs and MIS-HEMTs based on X-parameter analysis. From the X-parameter measurements, it can be seen that the output power, power gain, and linearity of the MIS-HEMT are much better than those of the HEMT because it has fewer surface traps. Therefore, traps on the AlGaN surface are the major concern for devices used in RF power applications and must be removed during fabrication process. Although the surface state can be removed by passivation technology, the traps in the GaN buffer still degrade the device power performance. In addition, the quality of the passivation layer must be taken into account to avoid excess traps induced in this film. Furthermore, we have used the X-parameter model to simulate the large-signal RF current waveform. It is more convenient than the equivalent circuit model for assessing the device large-signal performance. From the modeling result, serious current distortion in HEMTs without passivation was observed. In addition, the RF current gain in HEMTs was much lower than that in MIS-HEMTs with passivation. After suppressing the buffer-related trapping effect with UV illumination, the signal distortion and gain were improved.

## Figures and Tables

**Figure 1 micromachines-14-01011-f001:**
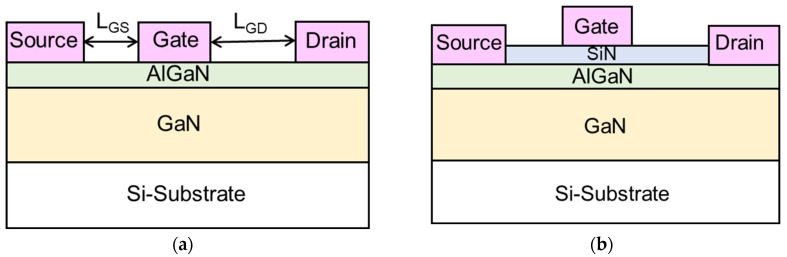
Schematic cross sections of (**a**) AlGaN/GaN HEMT and (**b**) AlGaN/GaN MIS-HEMT.

**Figure 2 micromachines-14-01011-f002:**
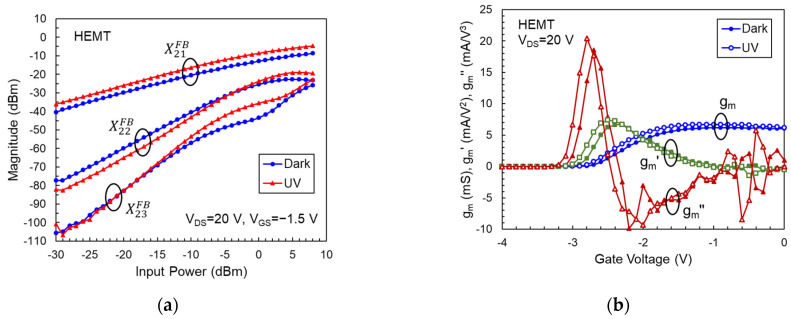
(**a**) Magnitudes of $X_{21}^{\mathrm{FB}}$, $X_{22}^{\mathrm{FB}}$, and $X_{23}^{\mathrm{FB}}$ versus input power for an AlGaN/GaN HEMT at V_GS_ = −1.5 V and V_DS_ = 20 V. (**b**) Magnitudes of g_m_, g_m_′, and g_m_″ versus gate voltage at V_DS_ = 20 V.

**Figure 3 micromachines-14-01011-f003:**
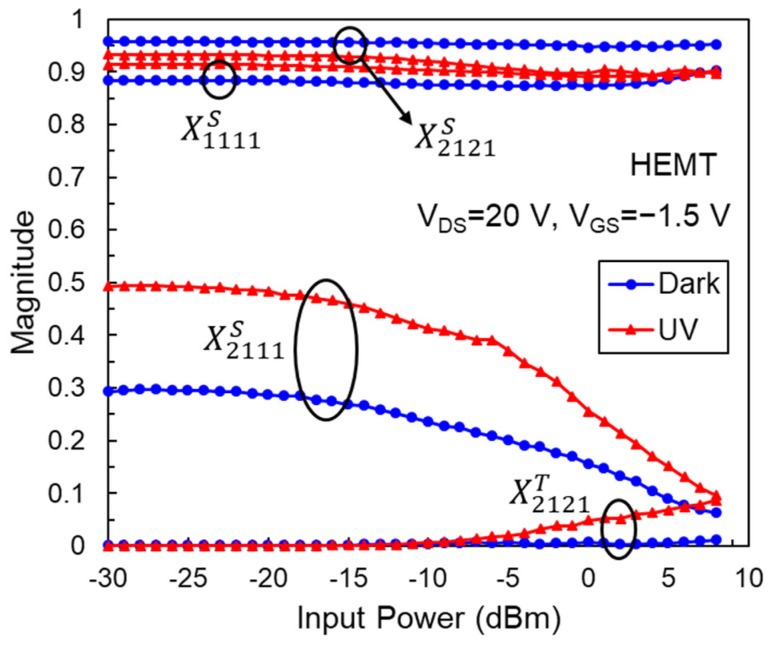
Magnitudes of $X_{1111}^{S}$, $X_{2111}^{S}$, $X_{2121}^{S}$, and $X_{2121}^{T}$ versus input power for an AlGaN/GaN HEMT.

**Figure 4 micromachines-14-01011-f004:**
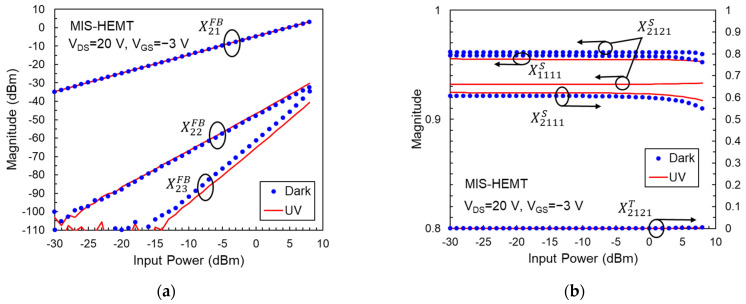
(**a**) Magnitudes of $X_{21}^{\mathrm{FB}}$, $X_{22}^{\mathrm{FB}}$, and $X_{23}^{\mathrm{FB}}$ versus input power for an AlGaN/GaN MIS-HEMT at V_GS_ = −3 V and V_DS_ = 20 V. (**b**) Magnitudes of $X_{1111}^{S}$, $X_{2111}^{S}$, $X_{2121}^{S}$, and $X_{2121}^{T}$ versus input power for this AlGaN/GaN MIS-HEMT.

**Figure 5 micromachines-14-01011-f005:**
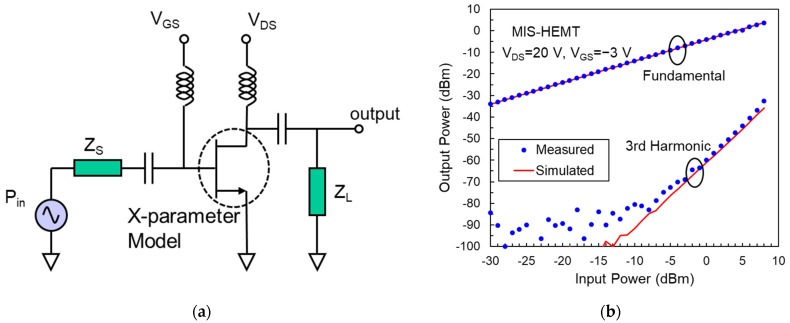
(**a**) Circuit configuration for device simulation using an X-parameter model. (**b**) Measured and simulated RF power characteristics of an AlGaN/GaN MIS-HEMT.

**Figure 6 micromachines-14-01011-f006:**
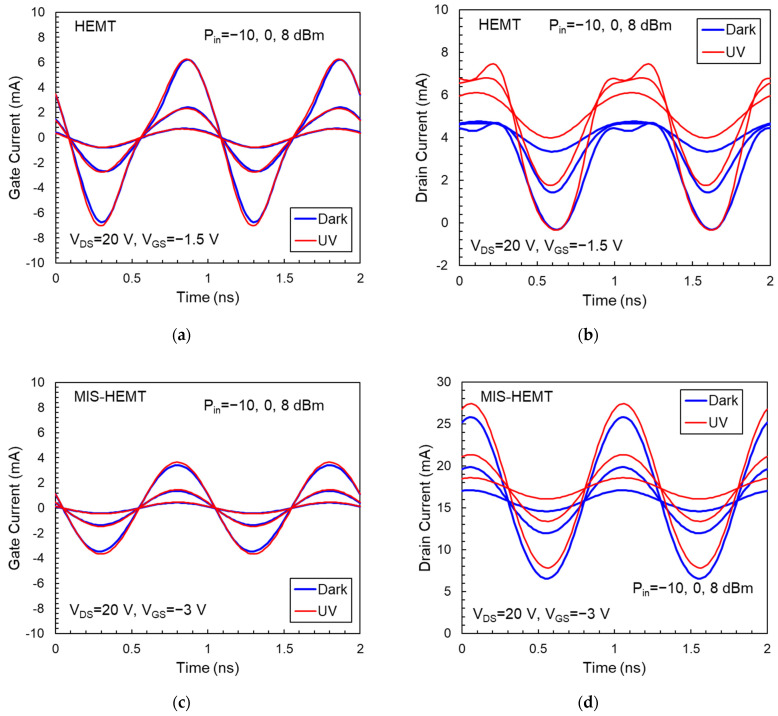
(**a**,**b**) Gate current and drain current waveforms of an AlGaN/GaN HEMT. (**c**,**d**) Gate current and drain current waveforms of an AlGaN/GaN MIS-HEMT.

## Data Availability

The data used to support the findings of this study are available within the article.

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
