# Peer review of "Analysis of Trapping Effect on Large-Signal Characteristics of GaN HEMTs Using X-Parameters and UV Illumination"

_micromachines, 2023, doi:10.3390/mi14051011_

Round 1

Reviewer 1 Report

In this article, the authors perform the extraction of a behavioral model of a GaN HEMT by means of the X-parameter approach under different trapping conditions. It is especially interesting that trapping is here excited by means of UV illumination, which possibly provides a better control of the trapping state of the device.

- In the Introduction, the authors mention that trapping is typically described by means of equivalent-circuit approaches, which is time-consuming. However, the authors should note that there exist many behavioral techniques that, despite being sometimes based on simple circuits or look-up tables, are very easy to simulate and feature an efficient/fast extraction based on measurements. Many of these make use of pulse-IV or double-pulse IV techniques. I would suggest to enlarge their literature search and discussion by including this type of models, see for example T. M. Martín-Guerrero, et al., "Automatic Extraction of Measurement-Based Large-Signal FET Models by Nonlinear Function Sampling," in IEEE Transactions on Microwave Theory and Techniques, 2020 and alike, which propose a similar multi-harmonic experiment like X-parameters and includes a straightforward trapping model.

- The authors should better explain the impact of UV on the trapping. What is the effect they aim to achieve in microelectronics terms? It seems from the results that UV enhanced de-trapping for a better performance, but could the author provide some more detailed explanation?

- The work provides interesting experimental data in the sense that it allows to analyze the waveforms under large-signal conditions for UV illumination. However, the measurement results should be better commented in relationship to the expected trapping from UV illumination.

Reviewer 2 Report

Chen et al. has reported Analysis of Trapping Effect on Large-Signal Characteristics of  GaN HEMTs Using X-Parameters and UV Illumination. The work is interesting and can be published after minor corrections.

1. What is the role of the dielectric layer in the current device structure? 

2. In the abstract, the authors has written X21 FB and X2111 it should be the full name then abbreviation.

3.  Figure 4a the data points are hiding with scale

Round 2

Reviewer 1 Report

Thanks for replying to the comments.